# Pneumothorax and/or Pneumomediastinum Worsens the Prognosis of COVID-19 Patients with Severe Acute Respiratory Failure: A Multicenter Retrospective Case-Control Study in the North-East of Italy

**DOI:** 10.3390/jcm10214835

**Published:** 2021-10-21

**Authors:** Matteo Bonato, Alessia Fraccaro, Nicholas Landini, Giuseppe Zanardi, Cosimo Catino, Francesca Savoia, Nicola Malacchini, Fabiola Zeraj, Piera Peditto, Vito Catalanotti, Elisabetta Marcon, Emanuela Rossi, Alessia Pauletti, Silvia Galvan, Riccardo Adami, Marta Tiepolo, Mauro Salasnich, Maria Cuzzola, Francesca Zampieri, Marcello Rattazzi, Mario Peta, Simonetta Baraldo, Marina Saetta, Giovanni Morana, Micaela Romagnoli

**Affiliations:** 1Pulmonology Unit, Ospedale Cà Foncello, Azienda Unità Locale Socio-Sanitaria 2 Marca Trevigiana, 31100 Treviso, Italy; alessa.fraccaro@aulss2.veneto.it (A.F.); giuseppe.zanardi@aulss2.veneto.it (G.Z.); cosimo.catino@aulss2.veneto.it (C.C.); francesca.savoia@aulss2.veneto.it (F.S.); nicola.malacchini@aulss2.veneto.it (N.M.); fabiola.zeraj@aulss2.veneto.it (F.Z.); piera.peditto@aulss2.veneto.it (P.P.); mauro.salasnich@aulss2.veneto.it (M.S.); maria.cuzzola@aulss2.veneto.it (M.C.); francesca.zampieri@aulss2.veneto.it (F.Z.); micaela.romagnoli@aulss2.veneto.it (M.R.); 2Department of Cardiac, Thoracic, Vascular Sciences and Public Health, University of Padova, 35121 Padova, Italy; simonetta.baraldo@unipd.it (S.B.); marina.saetta@unipd.it (M.S.); 3Department of Radiology, Ospedale Cà Foncello, Azienda Unità Locale Socio-Sanitaria 2 Marca Trevigiana, 31100 Treviso, Italy; nicholas.landini@aulss2.veneto.it (N.L.); giovanni.morana@aulss2.veneto.it (G.M.); 4IRCCS Azienda Ospedaliera-Universitaria, Division of Respiratory and Critical Care Sant’Orsola Hospital, 40141 Bologna, Italy; vito.catalanotti@aulss2.veneto.it; 5Pulmonology Unit, Ospedale di Vittorio Veneto, Azienda Unità Locale Socio-Sanitaria 2 Marca Trevigiana, 31029 Vittorio Veneto, Italy; elisabetta.marcon@aulss2.veneto.it; 6Pulmonology Unit, Ospedale di Montebelluna, Azienda Unità Locale Socio-Sanitaria 2 Marca Trevigiana, 31044 Montebelluna, Italy; emanuela.rossi@aulss2.veneto.it (E.R.); alessia.pauletti@aulss2.veneto.it (A.P.); 7Internal Medicine II, Ospedale Cà Foncello, Azienda Unità Locale Socio-Sanitaria 2 Marca Trevigiana, 31100 Treviso, Italy; silvia.galvan@aulss2.veneto.it; 8Geriatric Unit, Ospedale Cà Foncello, Azienda Unità Locale Socio-Sanitaria 2 Marca Trevigiana, 31100 Treviso, Italy; adami.riccardo@aulss2.veneto.it; 9Internal Medicine I, Ospedale Cà Foncello, Azienda Unità Locale Socio-Sanitaria 2 Marca Trevigiana, 31100 Treviso, Italy; marta.tiepolo@aulss2.veneto.it (M.T.); marcello.rattazzi@aulss2.veneto.it (M.R.); 10Intensive Care Unit, Ospedale Cà Foncello, Azienda Unità Locale Socio-Sanitaria 2 Marca Trevigiana, 31100 Treviso, Italy; mario.peta@aulss2.veneto.it

**Keywords:** case-control, coronavirus, SARS-CoV-2, pneumothorax, pneumomediastinum, COVID-19, ARDS

## Abstract

Pneumothorax (PNX) and pneumomediastinum (PNM) are potential complications of COVID-19, but their influence on patients’ outcomes remains unclear. The aim of the study was to assess incidence, risk factors, and outcomes of severe COVID-19 complicated with PNX/PNM. Methods: A retrospective multicenter case-control analysis was conducted in COVID-19 patients admitted for respiratory failure in intermediate care units of the Treviso area, Italy, from March 2020 to April 2021. Clinical characteristics and outcomes of patients with and without PNX/PNM were compared. Results: Among 1213 patients, PNX and/or PNM incidence was 4.5%. Among these, 42% had PNX and PNM, 33.5% only PNX, and 24.5% only PNM. COVID-19 patients with PNX/PNM showed higher in-hospital (*p* = 0.02) and 90-days mortality (*p* = 0.048), and longer hospitalization length (*p* = 0.002) than COVID-19 patients without PNX/PNM. At PNX/PNM occurrence, one-third of subjects was not mechanically ventilated, and the respiratory support was similar to the control group. PNX/PNM occurrence was associated with longer symptom length before hospital admission (*p* = 0.005) and lower levels of blood lymphocytes (*p* = 0.017). Conclusion: PNX/PNM are complications of COVID-19 associated with a worse prognosis in terms of mortality and length of hospitalization. Although they are more frequent in ventilated patients, they can occur in non-ventilated, suggesting that mechanisms other than barotrauma might contribute to their presentation.

## 1. Introduction

Coronavirus disease 2019 (COVID-19) caused by the new coronavirus 2 (SARS-CoV-2) spread in late 2019 in China and subsequently all over the world in 2020, with Italy being the first involved country in Europe at the end of February 2020. SARS-CoV-2 may lead to bilateral pneumonia and to severe acute respiratory failure. Up to 5 October 2021, 234,809,103 subjects have been infected, causing more than 4,800,345 deaths all over the world [1]. Pneumothorax (PNX) and pneumomediastinum (PNM) have been described as potential complications in some pulmonary infections [2], including the other two coronaviruses, causing SARS and MERS [3]. Several case series and studies, some of them included in a recent systematic review [4], investigated these complications in hospitalized COVID-19 patients, highlighting the occurrence of PNX/PNM also in COVID-19. However, their effective incidence and their influence on disease outcome are still debated [5,6,7,8]. A retrospective study reported an overall incidence of PNX/PNM of 0.54% in COVID-19 patients at emergency department admission [9], while an incidence of 6.1–15% has been reported by studies conducted in invasively ventilated patients [5,6,7,8,10]. These studies reported that COVID-19 is burdened by a higher incidence of PNX/PNM than the general population [6,9]. Since the pandemic outbreak in Italy (February 2020), the three pulmonology intermediate care units (IMCU) of Treviso area (Ca’ Foncello Hospital, via Ospedale 1, 31100 Treviso; Montebelluna City Hospital, via Togliatti 1, 31044 Montebelluna; and Vittorio Veneto City Hospital, via Forlanini 1, 31029 Vittorio Veneto), in the North-East of Italy (tot. 887.806 inhabitants), have treated in their intermediate care beds a total of 1213 patients affected by acute respiratory failure secondary to COVID-19.

The aim of our study is to assess among COVID-19 patients hospitalized for acute respiratory failure whether the occurrence of PNX/PNM was associated with a worse outcome in terms of mortality and length of hospitalization. Secondly, we aim to evaluate the incidence of these complications and possible risk factors for their presentation. 

## 2. Materials and Methods

### 2.1. Study Design and Participants

We retrospectively examined medical records of all subjects admitted to the pneumology intermediate care units of Treviso area (Veneto Region, North-East of Italy) of Ca’ Foncello Hospital (Treviso), Montebelluna City Hospital (Montebelluna), and Vittorio Veneto City Hospital (Vittorio Veneto) for acute respiratory failure secondary to COVID-19 from March 2020 to April 2021. The study was approved by our referral Ethics Committee (N. 793 CE/Marca del 9/4/2020). Inclusion criteria were limited to a diagnosis of acute respiratory failure (PaO_2_ < 60 mmHg in room air at arterial blood gas analysis) secondary to bilateral pneumonia due to SARS-CoV-2 infection, confirmed in all cases by a real-time reverse transcription-polymerase chain reaction from a nasopharyngeal swab. Among the whole cohort, we compared clinical characteristics, past medical history, and outcome of patients with occurrence of pneumothorax and/or pneumomediastinum during their hospitalization (cases) to a matched group of patients who did not develop these complications (controls). Diagnosis of PNX/PNM was confirmed with computed tomography (CT) scan or chest radiography. Controls were selected from all patients hospitalized for acute respiratory failure secondary to COVID-19 related bilateral pneumonia and during the same period of time. Controls have been randomly chosen and sex-age matched to cases, using a random number generator in a 1:2 (case/control) ratio.

### 2.2. Outcome Measures and Clinical Variables

Clinical details of the enrolled patients included demographics, past medical history, laboratory tests at patient arrival, chest radiography, clinical management, and outcome. Demographics, smoke habit, past medical history, and COVID-19 symptoms onset were extracted from the electronic patient registry. Among laboratory tests, we recorded blood platelets, lymphocytes and neutrophils (cell/mcL), C-reactive protein (mg/dL), D-dimer (ng/dL), and PaO_2_/FiO_2_ (P/F) ratio at pulmonology arrival. We defined ARDS following the Berlin Definition [11]. We included in the analysis the day of hospitalization when PNX/PMN occurred, its side, and the need for chest drainage. For PNX/PNM group, we considered the ongoing respiratory support when the event occurred while for the control group, the most intensive needed respiratory support during hospitalization. Outcome variables were the in-hospital mortality, mortality at 90 days from admission, and the length of hospitalization.

### 2.3. Patients’ Management and Respiratory Support

Patients were admitted to pulmonology intermediate care units in the presence of respiratory failure (PaO_2_ < 60 mmHg in RA). All patients included in the study received a pharmacological treatment based on the most updated national guidelines [12] and were initially supported with conventional or HFNC oxygen therapy or non-invasive ventilation. In particular, all patients received endovenous steroid treatment (methylprednisolone 40 mg/die or dexamethasone 6 mg/die) for 10 days, then tapered progressively halving the dose every 5 days until suspension. In case of further worsening of gas exchanges during hospitalization, patients were moved to ICU and underwent intubation and invasive ventilation, according to national guidelines [13]. Non-invasive and invasive mechanical ventilation parameters were similar for all patients, according to national guidelines: a pressure support ventilation modality with positive end-expiratory pressure (PEEP) of 10 cmH_2_O and pressure support (PS) of 8–12 cmH_2_O was used [13]. 

### 2.4. Statistical Analysis

Variables are presented with frequencies and percentages for categorical variables and as median [1st–3rd quartile] for continuous variables. The difference in explanatory variables was assessed using a Chi-squared test or Fisher test for dichotomous and categorical variables, a *t*-test for normally distributed continuous variables, and a Mann–Whitney U test for non-normal distributed continuous variables. Survival data were used to generate Kaplan–Meier curves with log-rank test analysis. A *p*-value lower than 0.05 was considered significant. Statistical analyses were performed with SPSS (IBM Corp. Released 2015. IBM SPSS Statistics for Windows, Version 23.0. Armonk, NY, USA: IBM).

## 3. Results

### 3.1. Incidence and Clinical Characteristics of Patients

Among 1213 patients admitted for acute respiratory failure due to COVID-19 between March 2020 and April 2021, a total of 53 (4.3%) were complicated by pneumothorax and/or pneumomediastinum during their hospitalization. The majority of cases were males (81.1%), with a median age of 69 years. 

The characteristics of the 53 patients with PNX/PNM are reported in Table 1. Twenty-two out of 53 (41.5%) had both PNX and PNM, 18 (33.5%) had PNX without PNM, and 13 (24.5%) had PNM alone. The most frequently affected side for PNX was the right one (52.5%), although five patients (12.5%) had bilateral PNX. In the majority of cases, PNX/PNM occurred during the hospitalization, with a median time lapse from admission of 9 [12–23.5] days. In some cases, PNX/PNM was already present at patient admission (17%) or incidentally reported on routine radiological exams during hospitalization in patients without suggestive symptoms (27% of cases). PNX/PNM was associated with subcutaneous emphysema in 54.7% of cases. In almost half (49.1%) of PNX, patients required treatment with a chest drainage, and two patients required chemical pleurodesis. Fourteen patients (26.4% of cases) died with persistent PNX/PNM, and one developed chronic pneumothorax. In others, the median radiological resolution length was 8 days. As shown in Table 1, at PNX/PNM occurrence, approximatively two-thirds of patients were ventilated (35.8% non-invasively and 32.1% invasively), while the remaining one-third of patients were oxygen-supported only: 15.1% with conventional oxygen support and 17% with high-flow nasal cannulas (HFNC). The majority of patients received methylprednisolone and minority dexamethasone without differences between the two groups.

### 3.2. Assessment of Risks Factors

Figure 1 compares the distribution of the respiratory support in progress at the time of PNX/PNM occurrence with the most intensive respiratory support during hospitalization in the control group. The distribution of the respiratory support was not different between the two groups. 

Table 2 shows past medical history and admission characteristics in patients with PNX/PNM and controls. Most of the patients had never smoked, and only a minority were current smokers, without differences between the two groups. Regarding respiratory and systemic comorbidities (e.g., hypertension, diabetes, obesity, dyslipidemia, chronic kidney disease, and heart disease), their prevalence did not significantly differ among the two groups. Patients in PNX/PNM group had a longer duration of COVID-19 symptoms before hospital admission, with a median time lapse of 7 (4.5–10) days, significantly higher than controls (5 (2–8) days; *p* = 0.005). Of note, no patient was vaccinated for COVID-19.

Among laboratory parameters at admission, PNX/PNM group showed a significant lower count of blood lymphocytes (670 (465–860) vs. 760 (540–1100) cell/mcl; *p* = 0.017) and a significant higher neutrophil-to-lymphocyte ratio (NL ratio) (9.7 (6.4–18.7) vs. 6.1 (4.2–12.8); *p* = 0.028) than controls. No significant differences were found for blood platelets, C-RP, D-dimer, or P/F ratio. Furthermore, the prevalence of ARDS was similar between the two groups (84.9% vs. 90.5%; *p* = n.s.).

### 3.3. Assessment of Outcome

Twenty-five out of the total fifty-three cases (47.2%) were deceased during hospitalization. In-hospital mortality was significantly higher in PNX/PNM patients than controls (47.2% vs. 27.4%; *p* = 0.02; OR (odds ratio) 2.37, 95%CI (confidence interval) 1.19–4.71; Figure 2a). Moreover, the occurrence of PNX/PNM significantly increased the length of hospitalization (25 (16–32.5) vs. 18 (10–27) days; *p* = 0.002; Figure 2b) compared to controls. At 90 days from admission follow-up, mortality rate was still higher in PNX/PNM group (47.2 vs. 31.1%; *p* = 0.048; OR 1.97 95%CI 1.002–3.89]. Figure 3 shows the Kaplan–Maier curve of survival at 90 days from admission of COVID-19 with and without PNX/PNM. Survival at 90 days was not significantly lower in PNX/PNM patients, although a trend was present (*p* = 0.07). Stratifying cases according to the occurrence of PNX + PNM (*n* = 22), PNX alone (*n* = 18), or PNM alone (*n* = 13), a similar in-hospital mortality rate (45.5% vs. 50% vs. 46.1%, respectively; *p* = n.s.), and a similar median length of hospitalization (23.5 vs. 27 vs. 27 days, respectively; *p* = n.s.) were observed. Among 53 cases, comparing males and females, there was no significant difference in terms of in-hospital mortality rate (51% vs. 30%; *p* = n.s.) or median length of hospitalization (25 vs. 27 days; *p* = n.s.). Considering a cut-off at 75 years of age (75th percentiles of age distribution in the whole cohort), in-hospital mortality and median length of hospitalization among the PNX/PNM group were similar between older and younger patients (46.6% vs. 47.3% and 24 vs. 27 days, respectively; *p* = n.s.). 

## 4. Discussion

Our study analyzed the clinical characteristics of COVID-19 patients complicated by pneumothorax and/or pneumomediastinum on a peculiar cohort of COVID-19 patients hospitalized in pulmonology intermediate care units for respiratory failure. It reported a worse prognosis in severe COVID-19 patients who were complicated with PNX/PNM at admission or during hospitalization, showing a prolonged hospitalization and an increased in-hospital and 90-day mortality. Moreover, in our study, these results have been shown to be independent of age, sex, and the severity of the disease based on the P/F ratio and the need for respiratory support. Many studies reported similar results in both invasively and non-invasively ventilated patients [7,8,10,14], although not all studies assessed the same outcomes [4,6,7]. In particular, Martinelli et al. [7] observed that the occurrence of PNX/PNM worsened the prognosis in older subjects, while we did not observe such age-related differences in our cohort. 

Although in the present analysis these two complications did not appear to be related to the disease severity, studies on selective cohorts of invasively ventilated patients in ICU reported higher incidences (6.1–15%) [4,5,6,8,14,15], while in all hospitalized COVID-19 patients, lower incidences were reported (0.97–2%) [4,7,8,9,16]. Our result (4.5%) ranks in the middle, probably mirroring the heterogeneity composition of our cohort. Moreover, this suggests that pneumothorax and pneumomediastinum in COVID-19 should not be considered rare events, especially considering the poorer prognosis associated with their occurrence. 

Concerning risk factors, we reported that the occurrence of PNX/PNM cannot be considered as related only to the severity of the acute respiratory failure and/or to the need for respiratory support. Indeed, in 17% of cases, these complications presented at hospital admission (spontaneous PNX/PNM), then before any treatment with oxygen and/or ventilation. Furthermore, during hospitalization, these two complications occurred mostly in non-invasively ventilated patients (35.8%) and in invasively ventilated patients (32.1%) but also in patients only treated with oxygen support (17% of patients treated with HFNC and 15.1% treated with conventional oxygen support). These findings support the speculation that barotrauma cannot be the only pathophysiological mechanism underlying PNX/PNM occurrence in these patients [6]. 

As previously mentioned [2], several lung infections have been reported to induce air leaks, leading to spontaneous pneumomediastinum or pneumothorax, including fungi, Mycoplasma pneumoniae, and Staphylococcus pneumonia, with viruses being a rare cause. The pathophysiological reasons might include the cough and/or the airway obstruction caused by endobronchial secretions, resulting in alveolar hyperinflation and increased endo-alveolar pressure [17]. This potential pathophysiological mechanism (Macklin effect) has been recently hypothesized also in COVID-19 [18]. 

The longer time lapse from symptoms onset to hospital admission observed in the PNX/PNM group of our study could reflect a prolonged length of respiratory distress on patients at home, which may result in more lung hyperinflation. We can also hypothesize that severe diffuse alveolar damage may be caused by immunological injury, resulting in interstitial emphysema and air tracking along the bronchoalveolar way to the mediastinum, leading to pneumothorax and pneumomediastinum. In line with this hypothesis, in our cohort, we observed that patients with PNX/PNM showed a reduced blood lymphocyte count at hospital admission, which is a well-known feature of SARS-CoV-2 infection with a recognized negative prognostic value [19]. As for influenza, SARS-CoV-2 infection may lead to uncontrolled lymphocytes traffic from peripheral blood to the lung, resulting in an increased immunological alveolar injury and, in turn, to an increased risk of pneumothorax/pneumomediastinum [20].

Moreover, similarly to controls, most of the patients presenting with PNX/PNM were non-smokers and did not present previous respiratory comorbidities, indicating that smoking history or prior pulmonary status do not appear to be involved in PNX/PNM pathogenesis. Considering these results, the possible etiology and physiopathology of PNX/PNM in COVID-19 patients remain challenging.

As a demonstration of the impact of these complications of COVID-19, numerous studies have been published on this topic since the pandemic outbreak, but only nine included a large number of patients, and only two included a control group (4). The results of a recently published study considering more than 130,000 hospitalized COVID-19 patients in the United Kingdom (8) are in line with the present study. However, there are some differences with previous works that have to be highlighted. Firstly, our results focused on the intermediate care context, which has not been considered in previous reports and may be a reference for this clinical setting. Moreover, it should be noted that the selection criteria of our patients were more restrictive, increasing the strength of our work: only patients with a certain diagnosis of COVID-19 performed by polymerase chain reaction were included, and we deliberately did not include patients with a presumptive diagnosis. Lastly, in our cohort, we could provide a more precise clinical characterization of patients, inclusive of laboratory data and time lapse before hospital admission, 90-day mortality, and length of hospitalization.

The main limitation of our study, as in most of the previously published studies, is its retrospective design, which does not allow us to ascribe a predictive value to the results on the outcome nor a cause–effect value to the risk factors. Another substantial limitation is the lack of the complete set of ventilation parameters, which were unfortunately not routinely registered in all patients. Notwithstanding this limitation, national guidelines for invasive and non-invasive ventilation were followed in the management of all patients; thus, we can assume that pressure differences between groups were minimal.

In conclusion, we demonstrated in a large cohort of severe COVID-19 patients hospitalized in pulmonology intermediate care units that pneumothorax and/or pneumomediastinum occurrence worsens the clinical outcome in terms of hospitalization length and mortality. Furthermore, we observed that pneumothorax and pneumomediastinum are associated with an increased time lapse between symptom onset and hospitalization and with reduced blood lymphocytes count. Finally, we reported that the respiratory support in our cohort is not closely associated with the occurrence of PNX/PNM. These findings support the evidence that multiple pathophysiological mechanisms other than barotrauma could concur in complicating the course of COVID-19 with PNX/PNM.

## Figures and Tables

**Figure 1 jcm-10-04835-f001:**
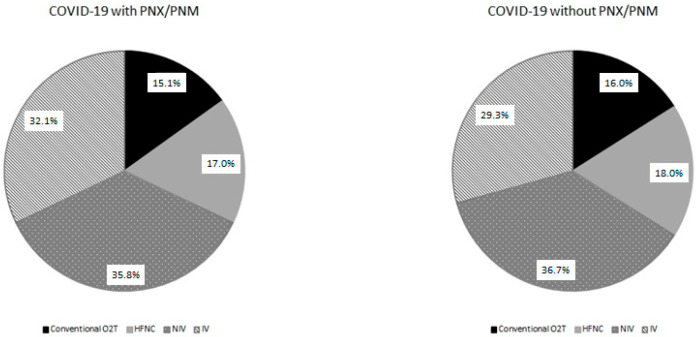
Pie chart reporting the distribution of respiratory support in COVID-19 patients with and without PNX/PNM. Relative percentages are reported in the labels.

**Figure 2 jcm-10-04835-f002:**
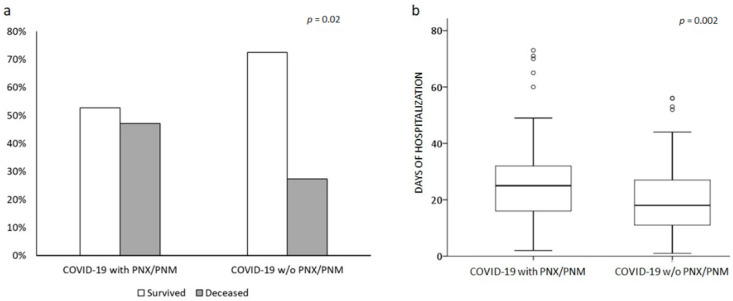
Comparison between COVID-19 patients with and w/o PNX/PNM. Bar line graph (**a**) compared the different in-hospital mortality rates (light grey deceased, dark grey survived) between the two groups. Box plot graph (**b**) compared the different lengths of hospitalization in the two groups. Solid line represents the median; bottom and top of the boxes are the 25th and 75th percentiles; brackets correspond to the 10th and the 90th percentiles.

**Figure 3 jcm-10-04835-f003:**
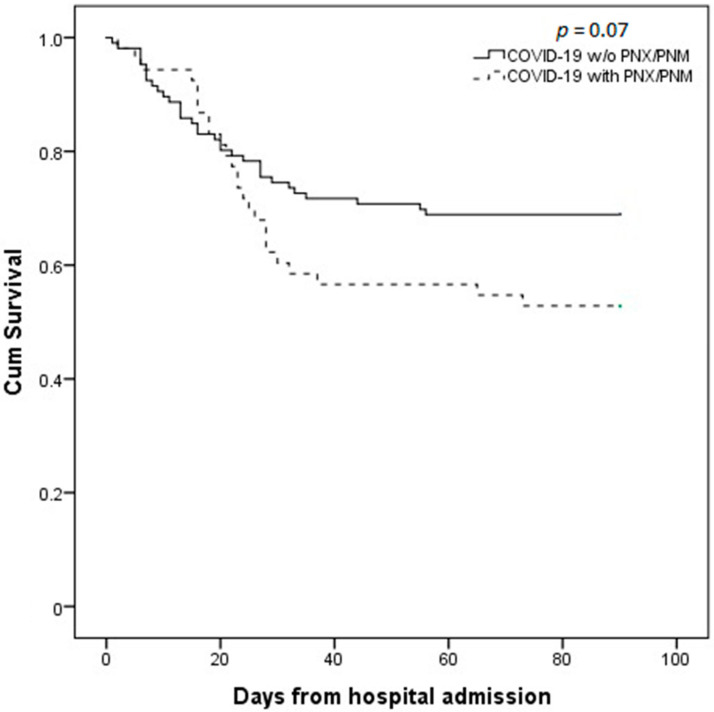
Kaplan–Meier survival curve comparison between patients who developed PNX/PNM and patients who did not, at 90 days from hospital admission. Continuous line represents COVID-19 without PNX/PNM, while dashed line represents COVID-19 patients with PNX/PNM. Survival curves are not statistically different, although a trend is observed (*p* = 0.07).

**Table 1 jcm-10-04835-t001:** Demographics, clinical characteristics, and outcome of COVID-19 patients with and without PNX/PNM.

	COVID-19 with PNX/PNM *n* = 53	COVID-19 w/o PNX/PNM *n* = 106	*p*-Value
Male, *n* (%)	43 (81.1%)	86 (81.1%)	n.s.
Age (y)	69 [62–75]	69 [62–75]	n.s.
Days to PNX/PNM occurrence		n.a.	
From symptoms onset	17 [12–23.5]		
From hospital admission	9 [4–16]		
PNX + PNM, *n* (%)	22 (41.5%)	n.a.	
PNX alone, *n* (%)	18 (33.5%)		
PNM alone, *n* (%)	13 (24.5%)		
Side of PNX, *n* (%)			
Right	21 (52.5%)		
Left	14 (35%)		
Bilateral	5 (12.5%)		
Prevalence of subcutaneous emphysema, *n* (%)	29 (54.7%)	n.a.	
Chest tube drainage treated, *n* (%)	26 (49.1%)	n.a.	
Median RX resolution length (days)	8 [5–13]	n.a.	
Respiratory support, *n* (%)			
Non-ventilated	17 (32.1%)	36 (34%)	
Conventional O2T	8 (15.1%)	17 (16%)	n.s.
HFNC	9 (17%)	19 (18%)	
Mechanically ventilated	36 (67.9%)	70 (66%)	
Non-invasively	19 (35.8%)	39 (36.7%)	
Invasively	17 (32.1%)	31 (29.3%)	
Endovenous steroid treatment			n.s.
Methylprednsiolone, *n* (%)	35 (66.1%)	76 (71.7%)	
Dexhametasone, *n* (%)	18 (33.9%)	30 (28.3%)	
Outcome			
Length of hospitalization in survived (days)	25 [16–32.5]	18 [10–27]	0.002
Deceased in-hospital, *n* (%)	25 (47.2%)	29 (27.4%)	0.02
Deceased at 90 days from admission, *n* (%)	25 (47.2%)	33 (31.1%)	0.048

Data are expressed as median (interquartile range) or frequency absolute (relative). *p*-value are assessed using Mann–Whitney U test, Chi-square, or Fisher’s exact tests as appropriate. n.s.—not significant; n.a.—not applicable. PNX: Pneumothorax; PNM: pneumomediastinum; HFNC: high-flow nasal cannulas.

**Table 2 jcm-10-04835-t002:** Past medical history and admission characteristics of COVID-19 patients with and without PNX/PNM.

	COVID-19 with PNX/PNM *n* = 53	COVID-19 w/o PNX/PNM *n* = 106	*p*-Value
Smoke habit, *n* (%)			
Never	30 (56.6%)	54 (50.9%)	
Former	19 (35.8%)	43 (40.6%)	n.s.
Current	4 (7.5%)	9 (8.5%)	
Respiratory comorbidities, *n* (%)			
Asthma	3 (5.7%)	3 (2.8%)	n.s.
COPD	3 (5.7%)	9 (8.5%)	n.s.
Chronic interstitial disease	3 (5.1%)	1 (0.9%)	n.s.
Other Comorbidities, *n* (%)			
Hypertension	25 (47.2%)	61 (57.5%)	n.s.
Obesity	6 (11.3%)	22 (20.8%)	n.s.
Type 2 diabetes	8 (15.1%)	24 (22.6%)	n.s.
Dyslipidemia	8 (15.1%)	30 (28.3%)	n.s.
Chronic kidney disease	2 (3.7%)	6 (5%)	n.s.
Chronic heart disease	10 (18.8%)	14 (22.6%)	n.s.
Laboratory test on admission, *n* (%)			
Platelets (×10^6^ cell/mcL)	216 [148–330]	237 [181–330]	n.s.
n.v. 140,000–440,000			
Lymphocytes (cell/mcL)	670 [465–860]	760 [540–1100]	0.017
n.v. 1000–4500			
Neutrophils (cell/mcL)	5910 [4670–8590]	6060 [3790–8000]	n.s.
n.v. 1800–8000			
NL ratio	9.7 [6.4–18.7]	6.1 [4.2–12.8]	0.028
n.v. 1.8			
C-RP (mg/dL)	9.7 [3.6–14.4]	8.6 [4.1–19.1]	n.s.
n.v. <0.5 mg/dL			
D-dimer (ng/mL)	1216 [755–1966]	856 [552–1561]	n.s.
n.v. <500 ng/dL			
P/F ratio (mmHg)	220 [132–291]	174 [109–257]	n.s.
n.v. >400 mmHg			
Days from symptoms onset to hospital admission	7 [4.5–10]	5 [2–8]	0.005
ARDS at admission, *n* (%)	45 (84.9%)	96 (90.5%)	n.s.

Data are expressed as median [interquartile range] or frequency absolute (relative). *p*-values are assessed using Mann–Whitney U test, Chi-square, or Fisher’s exact tests as appropriate. n.v.—normal values; n.s.—not significant. COPD: chronic obstructive pulmonary disease.

## Data Availability

The data that support the findings of this study are available from the corresponding author, [M.B.], upon reasonable request.

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
