# Peer review of "Pneumothorax and/or Pneumomediastinum Worsens the Prognosis of COVID-19 Patients with Severe Acute Respiratory Failure: A Multicenter Retrospective Case-Control Study in the North-East of Italy"

_jcm, 2021, doi:10.3390/jcm10214835_

Round 1
Reviewer 1 Report
Dear authors and editors, thank you for asking me to review this article.
There are now a number of case series describing pneumothorax and pneumomediastinum in Covid-19, and at first glance, this paper is not novel anymore. Regionally and nationally in the UK, I am aware of very large datasets such as ISARIC that have looked at this, and showed increased mortality with pnx and pnm. It is good that the authors had a control group.
So specific comments
- The influence on patient outcomes is well documented. It is clear that pnm and pnx have negative consequences and carry mortality. the authors should note that within the text, and say their study confirms what is already known.
- line 56 is very out of date now, the figures relate to May 2021
- why did the authors only include those with respiratory failure? we know that pnm and pnx can present without respiratory failure. this will have led to bias and thus the authors need to do a more wide ranging search of everyone with Covid and then who has pnm/pnx if they want to give an idea of the rate of incidence
- the results section is well written, but all this is again well known now (A case series of pneumothorax, pneumomediastinum and surgical emphysema in coronavirus disease 2019 (COVID-19) - Jackson - AME Surgical Journal (amegroups.com))
- this is not the first descriptor of poor prognosis in these patients-see reference 8- please modify the text
Many thanks again
Author Response
Dear authors and editors, thank you for asking me to review this article.
There are now a number of case series describing pneumothorax and pneumomediastinum in Covid-19, and at first glance, this paper is not novel anymore. Regionally and nationally in the UK, I am aware of very large datasets such as ISARIC that have looked at this, and showed increased mortality with pnx and pnm. It is good that the authors had a control group.
We thank the reviewer for his/her thoughtful review and his/her appreciation on the usage of a control group. By this study we aimed to assess the incidence, risk factors and outcome of these two COVID-19 complications based on “real-clinical” data from a peculiar hospital setting - intermediate care - in a peculiar kind of COVID-19 patients, e.g. with acute moderate-to-severe respiratory failure. During pandemic, the 3 Pulmonology intermediate care units of our area managed a significant number of Covid-19 patients with bilateral pneumonia and acute respiratory failure requiring oxygen support with high flow nasal cannulas (HFNC), and/or CPAP and/or NIV. Thus, the peculiarity of the present cohort is the inclusion of a homogeneous severe Covid-19 population, which can be considered in itself a novel valuable result, especially for Colleagues who work with critically ill patients as in our experience. We are well conscious that at present - October 2021 - our results in general might not represent a real novelty in this research field, as we pointed out in the manuscript (lines 63-69; 228-232; 275-279). However, it should be noted that the validation of results on independent cohorts represent one of the fundamental elements of evidence-based medicine. Despite this, we understand the point of view of the reviewer, so we refined the form of some sentences in the revised manuscript with the purpose to clarify the relative non-novelty of these results.
So specific comments
- The influence on patient outcomes is well documented. It is clear that pnm and pnx have negative consequences and carry mortality. the authors should note that within the text, and say their study confirms what is already known.
We thank the reviewer for this observation. This statement, which had been already reported in the original manuscript, has now been reinforced (line 228-232; 275-279).
- line 56 is very out of date now, the figures relate to May 2021
We thank the reviewer and we do apologize for this negligence, we provided the updated data (line 59-60).
- why did the authors only include those with respiratory failure? we know that pnm and pnx can present without respiratory failure. this will have led to bias and thus the authors need to do a more wide-ranging search of everyone with Covid and then who has pnm/pnx if they want to give an idea of the rate of incidence
We thank the reviewer for raising this issue. As expressed above, our purpose was to investigate the incidence, risk factors and outcome of these two complications, focusing on severe COVID-19 admitted to our Pulmonology intermediate care units. Though we understand the reviewer concern that this may lead to a selection bias, we have highlighted that the aim of our study was to investigate Covid-19 patients with respiratory failure (who represent patients hospitalized in our wards). Following the reviewer concern we discussed in the text (line 236-238) that our incidence is probably influenced by the special characteristics of our patients.
- the results section is well written, but all this is again well known now (A case series of pneumothorax, pneumomediastinum and surgical emphysema in coronavirus disease 2019 (COVID-19) - Jackson - AME Surgical Journal (amegroups.com))
Thanks for your comment. Our answer to this comment is similar to the previous ones.
- this is not the first descriptor of poor prognosis in these patients-see reference 8- please modify the text
We provided to modify the sentence (line 233-238) in order to avoid misunderstandings.
Reviewer 2 Report
In this original article, the authors described the characteristics and outcome in Covid-19 patients with pneumothorax and/or pneumomediastinum. The authors reported very interesting information, which have impact on the management for Covid-19 patients with severe acute respiratory failure. There are some comments that are meant to improve the quality of the manuscript.
- You compared between patients with PNX/PNM and randomly chosen patients without PNX/PNM. Why did you select the method? I feel non-chosen analysis is suitable.
- I want to know the detail for the steroid usage and vaccination.
- For patients with PNX, how long was the duration of air leakage or full lung expansion?
- For PNX underwent chest tube drainage, were there cases which needed additional treatments such as pleurodesis or surgery?
Author Response
In this original article, the authors described the characteristics and outcome in Covid-19 patients with pneumothorax and/or pneumomediastinum. The authors reported very interesting information, which have impact on the management for Covid-19 patients with severe acute respiratory failure. There are some comments that are meant to improve the quality of the manuscript.
We are grateful to the reviewer for the positive reply. We provided to improve the manuscript enriching it with the details kindly requested by the reviewer.
1.You compared between patients with PNX/PNM and randomly chosen patients without PNX/PNM. Why did you select the method? I feel non-chosen analysis is suitable.
We thank the reviewer for this question. We preferred to select controls from the cohort (nested case-control) firstly because of the rarity (less than 5%) of the studied event which can lead to statistical bias due to sample imbalance toward non-events. Secondly, we chose this method because was more cost-effectiveness.
- I want to know the detail for the steroid usage and vaccination.
We thank the reviewer for this opportunity to clarify details of our cohort. We included in the new version of the manuscript more details about steroid use (line 118-120; line 156-157; Table 1). None of our patients were vaccinated for COVID-19 (line 178).
- For patients with PNX, how long was the duration of air leakage or full lung expansion?
We thank the reviewer for this comment which improved our work. We included in the new version of the manuscript novel data about the length of pneumothorax/pneumomediastinum resolution in survived patients (line 150-152; Table 1).
- For PNX underwent chest tube drainage, were there cases which needed additional treatments such as pleurodesis or surgery?
Two patients underwent chemical pleurodesis. We provided to add this information in the revised version of the manuscript (line 150-152).
Round 2
Reviewer 1 Report
Dear authors, thank you for revising this article, I have no further concerns, and I am happy to recommend this for publication.
Reviewer 2 Report
Thank you for revision. This article has been corrected according to my suggestion.